# Joint Power Control and Resource Allocation with Rate Fairness Consideration for SWIPT-Based Cognitive Two-Way Relay Networks

**DOI:** 10.3390/s23177620

**Published:** 2023-09-02

**Authors:** Chunling Peng, Guozhong Wang, Huaping Liu

**Affiliations:** 1School of Electrical and Electronic Engineering, Chongqing University of Technology, Chongqing 400054, China; chunlingp@163.com; 2School of Communication Engineering, Chongqing College of Electronic Engineering, Chongqing 401331, China; 3School of Communication and Information Engineering, Chongqing University of Posts and Telecommunications, Chongqing 400065, China; 4School of Electrical and Electronic Engineering, Oregon State University, Corvallis, OR 97331, USA; huaping.liu@oregonstate.edu

**Keywords:** SWIPT, cognitive two-way relay, power splitting, joint resource allocation

## Abstract

This paper investigates the power control and resource allocation problem in a simultaneously wireless information and power transfer (SWIPT)-based cognitive two-way relay network, in which two secondary users exchange information through a power splitting (PS) energy harvesting (EH) cognitive relay node underlay in a primary network. To enhance the secondary networks’s transmission ability without detriment to the primary network, we formulate an optimization to maximize the minimum transmission rates of the cognitive users by jointly optimizing power allocation at the sources, the time allocation of transmission frames and power splitting at the relay, under the constraint that the transmission power of the cognitive network is set not to exceed the primary user interference threshold to ensure primary work performance. To efficiently solve this problem, a sub-optimal algorithm named the joint power control and resource allocation (JPCRA) scheme is proposed, in which we decouple the non-convex problem into convex problems and use alternative steps in the optimization algorithm to get final solutions. Numerical results reveal that the proposed scheme enhances transmission fairness and outperforms three traditional schemes.

## 1. Introduction

The demand for spectrum resources is increasing rapidly as wireless communication continues to develop. However, the limited spectrum resources severely restrict the further growth of communication capacity. Cognitive radio is an effective technology to enhance spectrum efficiency through the reasonable reuse of the authorized spectrum. The technology mainly includes three modes: spectrum interweave, spectrum overlay and spectrum underlay [1,2]. Compared with interweave and overlay, underlay is simple to implement because no spectrum sensing is needed, and it has a better ability to realize spectrum sharing [3]. With underlay, in order to ensure a high priority of the primary user’s (PU) access of the spectrum, the transmit power of the cognitive users must be kept under the interference tolerance threshold of the PU. Therefore, power control becomes a key issue in optimizing the overall performance of the cognitive network [4].

In [5], Lee et al. investigate transmit power control for an underlay cognitive radio network by using a deep learning method that determines its own transmit power based solely on its local channel state information (CSI). In [6], Sarvendranath et al. develop an optimal and novel joint antenna selection and power adaptation rule that minimizes the average symbol error probability of a secondary user that is subject to two practically well-motivated constraints. Hu et al. [7] propose two optimal power control schemes from the long-term and short-term perspectives for a cognitive low orbit satellite constellation with terrestrial networks, which aims to maximize the delay-limited capacity and minimize the outage probability, respectively. In [8], Chuang et al. propose a dynamic multiobjective approach for power and spectrum allocation in a cognitive-based environment and propose a dynamic resource allocation algorithm comprising a hybrid initialization method and feasible point generation mechanisms to solve the dynamic multiobjective optimization problem. In [9], two efficient and low-complexity power control strategies are proposed for an ambient backscatter-based spectrum-sharing network, and with the backscatter prominent, there is no need to estimate all users’ CSI.

The relay technique can expand the transmission distance and improve the transmission reliability of the system. Integrating relay and cognitive techniques can further improve the transmission performance of the system [10,11]. In [12], the closed-form expression of outage probability for a cognitive multi-hop relay network is derived over Rayleigh fading channels, and an optimization problem to minimize the outage probability of the cognitive relay network is formulated and solved. In [13], a novel decentralized scheduling technique is developed for the cognitive multi-user multi-relay network, which operates on an incremental relaying mechanism and derives the outage probability of the secondary network for both the decode-and-forward (DF) and amplify-and-forward (AF) strategies. The transmission performance of a two-way AF cognitive relay network considering the influence of the primary network is studied in [14], and closed-forms of outage probability and bit error rate are derived. In [15], Yang et al. propose a dynamic power transmission scheme for non-orthogonal multiple access (NOMA) cognitive relay networks and derive the closed-form expressions of outage probability and average sum rate. Zhong and Zhang investigate relay selection in a two-way full-duplex AF relay network [16] and derive the system outage probability and bit error rate. In [17], Poornima et al. investigate the energy efficiency and the spectral efficiency performance of multi-hop full duplex cognitive relay networks.

Relaying often causes energy consumption issues, and forcing idle users to use their own energy to help the relay is difficult. Therefore, the energy issue in cognitive relay networks has become a topic of substantial research interest [18]. Introducing radio frequency (RF) wireless energy harvesting (EH) technology into cognitive relay networks could potentially solve both the energy and spectrum problems, which has attracted significant attention in the academic communities [19]. In [20], He et al. derive and compare the outage probabilities of the primary network and the EH cognitive network under direct transmission, single-user cooperation and multi-user cooperation scenarios. In [21], Shome et al. investigate the error probability of an energy harvesting co-operative cognitive radio network with several relay selection criteria. Wang et al. [22] study an energy harvesting-based secure transmission scheme for cognitive multi-relay networks and analyze the average secrecy rate, the secondary secrecy outage probability and the ergodic secrecy rate. More recently, some researchers have begun to study the SWIPT protocols and resource allocation for cognitive AF two-way relay networks [23,24]. The optimization model of [23] aims to maximize the total transmission throughput of the system, and the authors propose an algorithm by optimizing the transmit power of sensor nodes. In [24], the approximate closed-form expression of minimizing the outage probability and throughput is taken as the optimization objective, and the closed-form solution of the optimal power control parameters and power partition ratio are obtained. Shukla et al. [25] evaluate the performance of the proposed SWIPT-enabled NOMA system by considering both the perfect and imperfect successive interference cancellation for the legitimate users over Nakagami-m fading in terms of outage probability, system throughput and energy efficiency.

However, to the best of our knowledge, much of the previous references that emphasize performance optimization for an energy harvesting cognitive two-way relay network focus on the AF transmission protocol. On the other hand, the fairness issue and the interference effects of the main network on the secondary network are seldom studied. In our previous studies [26], we have investigated the power allocation under power control for a simultaneous wireless information and power transfer (SWIPT)-based cognitive two-way relay network with rate fairness consideration. However, the time allocation and power splitting issue have not been considered yet. In this paper, we consider the two-way DF cognitive relay network and investigate the jointly-optimum design based on the PS energy harvesting protocol, which aims to maximize the minimum cognitive user transmission rate with rate fairness and power control consideration. We achieve this by jointly optimizing the power allocation at source nodes, the time allocation of frames and the power splitting ratio at the relay. The main contributions are summarized as follows.
(1)We develop a joint optimization scheme to maximize the minimum cognitive user transmission rate under rate fairness and power control consideration. The goal is to maximize the minimum cognitive user transmission performance through the joint optimization of time, power and power component ratio.(2)A stepped alternating optimization algorithm is proposed to solve the complex non-convex optimization problem. Through decoupling, the original problem is transformed into convex optimization problems and an alternating optimization problem. This avoids solving the complex non-convex optimization problem.(3)The results show that the proposed scheme improves the unfairness of inter-user transmission caused by channel asymmetry, and its superiority over the traditional scheme in terms of outage probability is depicted.

The rest of this paper is organized as follows: Section 2 presents the system model and problem formulation. Section 3 proposes the joint power control and resource allocation scheme. Section 4 studies and compares the performance of the proposed scheme under a simulation system setup. Finally, Section 5 concludes the paper.

## 2. System Model and Problem Formulation

Consider a half-duplex two-way cognitive relay network that consists of two source nodes $S_{1}$ and $S_{2}$ with a fixed power supply and a passive relay node with energy harvesting ability. All the terminals are equipped with a single omnidirectional antenna, and the antenna gain is normalized to 1. It is assumed that a direct link between the source nodes does not exist. We adopt the power splitting receiver architecture and DF protocol at relay. The system model is as shown in Figure 1.

The information exchange of the whole transmission needs two time slots: a multiple access (MA) transmission phase and a broadcast (BC) phase. During the MA phase, source nodes $S_{1}$ and $S_{2}$ transmit their own information to relay *R*. Due to the broadcasting nature, PU receives the information from source nodes $S_{1}$ and $S_{2}$ as interference. To ensure the performance of the primary network, an interference threshold to restrict the total transmission power of source nodes $S_{1}$ and $S_{2}$ is set. Once the relay receives the signal, it partitions it into two parts: one part for energy harvesting, the other for information decoding. In the BC phase, the relay forwards the decoded signals to source nodes $S_{1}$ and $S_{2}$ with the harvested energy. Similarly, to ensure the performance of the primary network, the transmission power of the relay should be under the interference threshold.

### 2.1. Information and Energy Transfer

Let the total time of the whole transmission phase be normalized to be 1; if the MA phase time period is *t*, then the BC phase time period is 1−t. In the MA phase, the transmit power of each source node is $P_{i}$. Due to the peak power constraint, $P_{i}$ should satisfied the following equation:(1)$$\begin{array}{c} P_{i}\leq P_{i,\max},i=1,2 \end{array}$$
where $P_{i,\max}$ is the peak power of source node $S_{i}$.

Since the cognitive network utilizes underlay spectrum sharing, the received interference power for the primary user should be less than an interference threshold *Q* to satisfy the primary performance, i.e.,
(2)$$\begin{array}{c} \sum_{i=1}^{2}{\left|l_{i}\right|}^{2}P_{i}\leq Q,i=1,2 \end{array}$$
where $l_{i}$ is the CSI from source $S_{i}$ to PU, and $|l_{i}{|}^{2}P_{i},i=1,2$ is the interference caused by spectrum sharing from source node $S_{i}$ to PU.

Let α(0<α<1) be the interference constraint ratio of two cognitive sources to the primary network. The restrictions of $P_{1}$ and $P_{2}$ can be reformulated as follows.
(3)$$\begin{array}{cc} \; & P_{1}\leq \min\left(P_{1,\max},\frac{\alpha Q}{|l_{1}{|}^{2}}\right) \end{array}$$
(4)$$\begin{array}{c} P_{2}\leq \min\left(P_{2,\max},\frac{(1-\alpha )Q}{|l_{2}{|}^{2}}\right) \end{array}$$

The signal received at the cognitive relay is
(5)$$\begin{array}{c} y_{R}=h_{1}\sqrt{P_{1}}x_{1}+h_{2}\sqrt{P_{2}}x_{2}+\tau_{u,r}+n_{r,a} \end{array}$$
where $\tau_{u,r}∼\mathrm{CN}\left(0,\sigma_{u,r}^{2}\right)$ is the interference introduced by the primary network, $h_{i}$ is the CSI between source node $S_{i}$ and the relay *R*, and $n_{r,a}∼\mathrm{CN}\left(0,\sigma_{a}^{2}\right)$ is the white noise at the receiver.

The cognitive relay then splits the received signal $y_{R}$ into two parts: $\sqrt{\rho }y_{R}$ for energy harvesting and $\sqrt{1-\rho }y_{R}$ for information decoding. With linear energy harvesting, the harvested energy can be expressed as
(6)$$\begin{array}{c} E=\eta \rho (|h_{1}{|}^{2}P_{1}+|h_{2}{|}^{2}P_{2}+\sigma_{u,r}^{2}+\sigma_{a}^{2})\cdot t \end{array}$$
where 0<η<1 is energy conversion efficiency. Because for practical cases the noise power is far less than the signal power, we neglect white noise for simplicity of analysis. Thus, Equation (Equation 6) is written as
(7)$$\begin{array}{c} E=\eta \rho (|h_{1}{|}^{2}P_{1}+|h_{2}{|}^{2}P_{2}+\sigma_{u,r}^{2})\cdot t \end{array}$$

The signal used to decode information is written as
(8)$$\begin{array}{c} y_{ID}=\sqrt{1-\rho }\left(h_{1}\sqrt{P_{1}}x_{1}+h_{2}\sqrt{P_{2}}x_{2}+\tau_{u,r}+n_{r,a}\right)+n_{r,b} \end{array}$$
where $n_{r,b}∼\mathrm{CN}\left(0,\sigma_{b}^{2}\right)$ is the noise generated by signal conversion from band-pass to baseband [27]. Since $\sigma_{a}^{2}\ll \sigma_{b}^{2}$, in practice, for simplicity, we neglect $n_{r,a}$ in the following analysis.

According to Equation (Equation 8) and [10], the rate region of the MA phase is obtained as
(9)$$C_{MA}=\left\{\left(R_{1},R_{2}\right):\left\{\begin{array}{cc} & R_{1}\leq t\cdot C\left(\Upsilon_{1r}\right)\\ & R_{2}\leq t\cdot C\left(\Upsilon_{2r}\right)\\ & R_{1}+R_{2}\leq t\cdot C\left(\Upsilon_{MA}\right) \end{array}\right.\right\}$$
where $C\left(x\right)=\log_{2}\left(1+x\right)$, $\Upsilon_{ir},i=1,2$ is the signal to interference plus noise power ratio (SINR) from source $S_{i}$ to relay *R*, $\Upsilon_{MA}$ is the SINR of multiple access transmission, and
(10)$$\begin{array}{cc} \; & \Upsilon_{1r}=\frac{\left(1-\rho \right)\gamma_{1}}{\left(1-\rho \right)\sigma_{u,r}^{2}+\sigma_{b}^{2}} \end{array}$$
(11)$$\begin{array}{c} \Upsilon_{2r}=\frac{\left(1-\rho \right)\gamma_{2}}{\left(1-\rho \right)\sigma_{u,r}^{2}+\sigma_{b}^{2}} \end{array}$$
(12)$$\begin{array}{c} \Upsilon_{MA}=\frac{\left(1-\rho \right)\gamma_{\Sigma }}{\left(1-\rho \right)\sigma_{u,r}^{2}+\sigma_{b}^{2}} \end{array}$$
where $\gamma_{1}={\left|h_{1}\right|}^{2}P_{1}$, $\gamma_{2}={\left|h_{2}\right|}^{2}P_{2}$ and $\gamma_{\Sigma }=|h_{1}{|}^{2}P_{1}+{\left|h_{2}\right|}^{2}P_{2}$.

In the broadcast phase, the cognitive relay performs information decoding utilizing the harvested energy. Assuming perfect CSI, the relay can utilize physical layer network coding to encode the received signal $y_{ID}$ into $x_{R}=x_{1}⊕x_{2}$. Because of underlay spectrum sharing, the transmit power of relay *R* is restricted not only by the harvested energy but also by the interference threshold of PU, which is written as
(13)$$\begin{array}{cc} P_{R}\leq \frac{E}{1-t} & ={\eta \rho (|h_{1}|}^{2}P_{1}{+|h_{2}|}^{2}P_{2}+\sigma_{u,r}^{2})\frac{t}{1-t} \end{array}$$
(14)$$\begin{array}{cc} \; & \;\;\;\;|l_{r}{|}^{2}P_{R}\leq Q \end{array}$$

Equation (Equation 13) can be rewritten in a simpler form as
(15)$$\begin{array}{c} P_{R}\leq \min\left(\eta \rho \left(\gamma_{\Sigma }+\sigma_{u,r}^{2}\right)\frac{t}{1-t},\frac{Q}{|l_{r}{|}^{2}}\right). \end{array}$$

The signal received at source node $S_{i}$ is
(16)$$\begin{array}{c} y_{S_{i}}=h_{i}\sqrt{P_{R}}x_{R}+\tau_{u,i}+n_{i} \end{array}$$
where $\tau_{u,i}∼\mathrm{CN}\left(0,\sigma_{u,i}^{2}\right)$ is the interference caused by the primary network, $n_{i}∼\mathrm{CN}\left(0,\sigma^{2}\right)$ is white noise at source node $S_{i}$.

Source node $S_{i}$ decodes $x_{R}$ and then uses self-cancellation to decode the intended information. For example, source node $S_{1}$ decodes $x_{2}$: $x_{2}=x_{R}⊕x_{1}$. According to Equation (Equation 16) and [10], the rate region of the BC phase is obtained as
(17)$$C_{BC}=\left\{\left(R_{1},R_{2}\right):\left\{\begin{array}{cc} & R_{1}\leq \left(1-t\right)\cdot C\left(\Upsilon_{r2}\right)\\ & R_{2}\leq \left(1-t\right)\cdot C\left(\Upsilon_{r1}\right) \end{array}\right.\right\}$$
where $\Upsilon_{r1}$ and $\Upsilon_{r2}$ are the SINRs from the relay to $S_{1}$ and $S_{2}$, respectively, and are written as
(18)$$\begin{array}{cc} \; & \Upsilon_{r1}=\frac{|h_{1}{|}^{2}P_{R}}{\sigma_{u,i}^{2}+\sigma_{1}^{2}} \end{array}$$
(19)$$\begin{array}{c} \Upsilon_{r2}=\frac{|h_{2}{|}^{2}P_{R}}{\sigma_{u,i}^{2}+\sigma_{2}^{2}} \end{array}$$

### 2.2. Max-Min Optimization Problem Formulation

The goal is to assess the system’s potential transmission capability with fairness consideration for the cognitive SWIPT-based relay network. The primary network’s transmission should be guaranteed first. To this end, we propose joint power control and resource allocation optimization, aiming at maximizing the minimum transmission rate of the cognitive relay system. The optimization problem is formulated as
(20)$$\begin{array}{cc} \mathrm{OP}1:\;\;\; & \max\min(R_{1},R_{2})\\ s.t.\;\; & C1:(3a),(3b)\\ & C2:\mathrm{Equation}\;(15)\\ & C3:\mathrm{Equation}\;(9)\\ & C4:\mathrm{Equation}\;(17)\\ & C5:t\in (0,1)\\ & C6:\rho \in (0,1)\\ & C7:\alpha \in (0,1) \end{array}$$
where $P=\{P_{1},P_{2},P_{R}\}$, $C_{1}$ and $C_{2}$ are, respectively, the transmission power limits of the source nodes and relay; $C_{3}$ and $C_{4}$ are the transmission rate region limits of the MA phase and the BC phase, respectively; and C5, C6 and C7 are the range of the time allocation parameter, the range of the power splitting parameter and the range of the power control parameter, respectively.

Since multiple variables are coupled in conditions C2∼C4, OP1 is non-convex. Analysis reveals that when $P_{1}$ and $P_{2}$ are fixed, OP1 degenerates to a joint optimization problem determined by *t* and ρ; when *t* and ρ are fixed, OP1 degenerates to an optimization problem determined by α. Thus, the original problem can be decoupled into two parts deriving the optimal power control and power allocation parameters when *t* and ρ are fixed and deriving the time allocation and power splitting parameters with joint resource allocation when the transmission power is fixed. Based on the analysis, we develop a sub-optimal algorithm to solve this complex problem.

## 3. Joint Power Control and Resource Allocation

A sub-optimal algorithm to solve OP1 is proposed, which is named joint power control and resource (JPCRA) allocation. It is based on solving two degenerated optimization works: power allocation with power control (PAPC) consideration and jointly optimum time allocation and power splitting ratio (JoTAPS) with a fixed transmit power. The final results can be obtained by using the alternative optimization algorithm based on PAPC and JoTAPS. This technique is described in detail next.

### 3.1. Power Allocation with Power Control Consideration

One degenerated optimization work is proposing a power allocation scheme that considers power control, i.e., deriving the optimal power control and power allocation parameters when *t* and ρ are fixed. Equations (Equation 9) and (Equation 17) show that as $P_{1}$ or $P_{2}$ increases, the achievable upper bound of the objection function $\min(R_{1},R_{2})$ in the MA phase increases monotonically. As $P_{R}$ increases, $\min(R_{1},R_{2})$ in the BC phase increases monotonically. Thus, the system achieves optimal transmission performance with the maximum attainable values of $P_{1}$, $P_{2}$ and $P_{R}$ expressed as
(21)$$\begin{array}{cc} \; & P_{1}=\min\left(P_{1,\max},\frac{\alpha Q}{|l_{1}{|}^{2}}\right) \end{array}$$
(22)$$\begin{array}{cc} \; & P_{2}=\min\left(P_{2,\max},\frac{(1-\alpha )Q}{|l_{2}{|}^{2}}\right) \end{array}$$
(23)$$\begin{array}{cc} \; & P_{R}=\min\left(\eta \rho \left(\gamma_{\Sigma }+\sigma_{u,r}^{2}\right)\frac{t}{1-t},\frac{Q}{|l_{r}{|}^{2}}\right) \end{array}$$

Equations (Equation 21)–(Equation 23) show that $P_{1}$, $P_{2}$ and $P_{R}$ depend on α(0<α<1). When the sources have no transmit power limits, $P_{1}$, $P_{2}$ and $P_{R}$ can be rewritten as
(24)$$\begin{array}{cc} P_{1}= & \alpha Q/|l_{1}{|}^{2} \end{array}$$
(25)$$\begin{array}{cc} P_{2}= & \left(1-\alpha \right)Q/|l_{2}{|}^{2} \end{array}$$
(26)$$\begin{array}{cc} P_{R}= & \min\left(\eta \rho \left(\left(\frac{|h_{1}{|}^{2}\alpha }{|l_{1}{|}^{2}}+{\frac{|h_{2}{|}^{2}\left(1-\alpha \right)}{l_{2}}}^{2}\right)Q+\right.\right.\\ \; & \;\;\;\sigma_{u,r}^{2})\frac{t}{1-t},\frac{Q}{|l_{r}{|}^{2}}) \end{array}$$

By substituting Equations (24)–(26) into OP1, the optimization problem reduces to a one-dimensional optimization problem. Define $R_{\min}=\min\left(R_{1},R_{2}\right)$. This one-dimensional optimization problem can be expressed as
(27)$$\begin{array}{cc} \mathrm{OP}2: & \max\;\;R_{\min}\\ s.t.\; & {C3}^{\prime }:R_{\min}\leq t\cdot \min\left(C\left(\Upsilon_{ir}\right),C\left(\Upsilon_{MA}\right)/2\right)),i=1,2\\ & {C4}^{\prime }:R_{\min}\leq \left(1-t\right)\cdot C\left(\Upsilon_{ri}\right),i=1,2 \end{array}$$
where
(28)$$\begin{array}{cc} C(\Upsilon_{1r}) & =\log_{2}\left(1+\psi H_{1}\alpha \right) \end{array}$$
(29)$$\begin{array}{cc} C(\Upsilon_{2r}) & =\log_{2}\left(1+\psi H_{2}\left(1-\alpha \right)\right) \end{array}$$
(30)$$\begin{array}{cc} C(\Upsilon_{MA}) & =\log_{2}\left(1+\psi \left(H_{1}\alpha +H_{2}\left(1-\alpha \right)\right)\right) \end{array}$$
(31)$$\begin{array}{cc} C(\Upsilon_{ri}) & =\log_{2}\left(1+\frac{|h_{1}{|}^{2}P_{R}}{\sigma_{u,i}^{2}+\sigma_{1}^{2}}\right) \end{array}$$
and
(32)$$\begin{array}{cc} P_{R}\; & =\;\min\;\left(\;\eta \rho \left(\;\;\left(\;\frac{|h_{1}{|}^{2}\alpha }{|l_{1}{|}^{2}}\;+\;\frac{|h_{2}{|}^{2}\left(1\;-\;\alpha \right)}{|l_{2}{|}^{2}}\;\right)\;Q\;+\;\sigma_{u,r}^{2}\;\right)\;\;\frac{t}{1\;-\;t},\;\frac{Q}{|l_{r}{|}^{2}}\;\right) \end{array}$$
(33)$$\begin{array}{cc} H_{i} & =\frac{|h_{i}{|}^{2}}{|l_{i}{|}^{2}},i=1,2 \end{array}$$
(34)$$\begin{array}{cc} \psi & =\frac{(1-\rho )Q}{\left(\left(1-\rho \right)\sigma_{u,r}^{2}+\sigma_{b}^{2}\right)} \end{array}$$

Once the upper-bound of ${C3}^{\prime }$ and ${C4}^{\prime }$ as well as the intersection of ${C3}^{\prime }$ and ${C4}^{\prime }$ are determined, the optimal value of OP2 can be obtained by $R_{\min}=\min\left(R_{1},R_{2}\right)$. Thus, the first step is to find α, which maximizes $t\cdot \min(C\left(\Upsilon_{ir}\right),C\left(\Upsilon_{MA}\right)/2)$ and $\left(1-t\right)\cdot C\left(\Upsilon_{ri}\right)$. Through some mathematical analysis, OP2 can be solved in the following two cases by comparing the two terms of $P_{R}$ expressed in Equation (Equation 32).

(1) Case 1: When ${\left|l_{2}\right|}^{2}(Q-\sigma_{u,r}^{2}|l_{r}{|}^{2})-\eta \rho Q{\left|l_{r}\right|}^{2}{\left|h_{2}\right|}^{2}\leq 0$

In this case, $P_{R}=Q/{\left|l_{r}\right|}^{2}$, the upper bound of ${C4}^{\prime }$ is a constant that is not affected by α; the upper bound of ${C3}^{\prime }$ is a continuous piecewise function of α. The element $C(\Upsilon_{1r})$ in ${C3}^{\prime }$ is a monotonically-increasing function of α, and the element $C(\Upsilon_{2r})$ in ${C3}^{\prime }$ is a monotonically-decreasing function of α. $C(\Upsilon_{MA})$ is a monotonic function of α whose monotony is affected by the value of $H_{1}-H_{2}$. Denote $\alpha_{ir,ma}$ as the intersection of $C(\Upsilon_{ir})$ and $C(\Upsilon_{MA})/2$, $\alpha_{1r,2r}$ as the intersection of $C(\Upsilon_{1r})$ and $C(\gamma_{2r})$. The obtained α that maximizes $t\cdot \min(C\left(\Upsilon_{ir}\right),C\left(\Upsilon_{MA}\right)/2)$ is
(35)$$\alpha_{1}=\left\{\begin{array}{cc} & \alpha_{1r,2r},\;\mathrm{if}\;\alpha_{1r,ma}\geq \alpha_{2r,ma}\\ \; & \alpha_{1r,ma},\;\mathrm{if}\;\alpha_{1r,ma}<\alpha_{2r,ma}\;\mathrm{and}\;H_{1}>H_{2}\\ \; & \alpha_{2r,ma},\;\mathrm{if}\;\alpha_{1r,ma}<\alpha_{2r,ma}\;\mathrm{and}\;H_{1}\leq H_{2} \end{array}\right.$$

Since the upper bound of $C4^{\prime }$ is not affected by α, $\alpha_{1}$ is a feasible solution for maximizing the upper bound of $C4^{\prime }$ and even $R_{\min}$. Therefore, the optimal power control parameter of OP2 in Case 1 is $\alpha_{c1}^{*}=\alpha_{1}$.

(2) Case 2: When $|l_{2}{|}^{2}(Q-\sigma_{u,r}^{2}|l_{r}{|}^{2})-\eta \rho Q|l_{r}{|}^{2}{\left|h_{2}\right|}^{2}>0$

In this case, $P_{R}=\eta \rho (((|h_{1}{|}^{2}\alpha /|l_{1}{|}^{2}+|h_{2}{|}^{2}\left(1-\alpha \right)/|l_{2}{|}^{2})Q)+\sigma_{u,r}^{2})\frac{t}{1-t}$. Substituting $P_{R}$ into OP2, we have
(36)$$\begin{array}{cc} \mathrm{OP}3:\; & \max\;R_{\min}\\ s.t.\; & \;C3^{\prime }:R_{\min}\leq t\cdot \min\left(C\left(\Upsilon_{ir}\right),C\left(\Upsilon_{MA}\right)/2\right),i=1,2\\ & \;C2^{\prime \prime }:\;\alpha \leq \alpha_{0}\\ & \;C4^{\prime \prime }:\;R_{\min}\leq \left(1-t\right)\cdot C\left(\Upsilon_{ri}\right),i=1,2 \end{array}$$
where
(37)$$\begin{array}{cc} \alpha_{0}= & \frac{|l_{1}{|}^{2}|l_{2}{|}^{2}(Q-\sigma_{u,r}^{2}|l_{r}{|}^{2})}{\eta \rho Q|l_{r}{|}^{2}\left(\right|h_{1}{|}^{2}|l_{2}{|}^{2}-|h_{2}{|}^{2}|l_{1}{|}^{2})}\\ \; & -\frac{|h_{2}{|}^{2}{\left|l_{1}\right|}^{2}}{|h_{1}{|}^{2}|l_{2}{|}^{2}-|h_{2}{|}^{2}{\left|l_{1}\right|}^{2}} \end{array}$$
(38)$$\begin{array}{cc} C(\Upsilon_{ri})= & \log_{2}\left(1+\varphi_{i}\left(\left(H_{1}\alpha +H_{2}\left(1-\alpha \right)\right)Q+\sigma_{u,r}^{2}\right)\right) \end{array}$$
(39)$$\begin{array}{cc} \varphi_{i}= & \frac{|h_{i}{|}^{2}\eta \rho t}{\sigma_{u,i}^{2}+\sigma_{i}^{2}}\left(1-t\right),i=1,2 \end{array}$$

In this case, the upper bound of $C4^{\prime \prime }$ is a monotonically increasing function of α, which is affected by the value of $H_{1}-H_{2}$; the upper bound of $C3^{\prime \prime }$ is a continuous piecewise function of α, in which $C(\Upsilon_{1r})$ is a monotonically increasing function of α while $C(\Upsilon_{2r})$ is a monotonically decreasing function, and $C(\Upsilon_{MA})$ is a monotonic function of α affected by the value of $H_{1}-H_{2}$. Denote $\alpha_{ir,ma}$ as the intersection of $C(\Upsilon_{ir})$ and $C(\Upsilon_{MA})/2$, $\alpha_{1r,2r}$ as the intersection of $C(\Upsilon_{1r})$ and $C(\Upsilon_{2r})$ and $\alpha_{ir,rj},i,j=1,2$ as the intersection of $C(\Upsilon_{ir})$ and $C(\Upsilon_{ir})$. Some analysis leads to the following observations:(a)When $H_{1}=H_{2}$In this case, $C(\Upsilon_{MA})$ and $C(\Upsilon_{ri}^{\prime })$ are constants. Thus, OP3 has one and only one optimal α, $\alpha_{1}^{\prime }=\min\left(\alpha_{0},\alpha_{1r,2r}\right)$.(b)When $H_{1}>H2$In this case, $C(\Upsilon_{MA})$ and $C(\Upsilon_{ri}^{\prime })$ become monotonically increasing functions of α. By analyzing the relationship of the intersection, can obtain that
(40)$$\alpha_{2}^{\prime }=\left\{\begin{array}{cc} \; & \min\left(\alpha_{0},\alpha_{1r},\alpha_{2r}\right),\;\;\;\mathrm{when}\;\alpha_{\min,1}\geq \alpha_{\max,2}\\ & \min\left(\alpha_{0},\alpha_{\min,1}\right),\;\;\;\;\;\mathrm{when}\;\alpha_{\min,1}<\alpha_{\max,2} \end{array}\right.$$(c)When $H_{1}<H_{2}$In this case, $C(\Upsilon_{MA})$ and $C(\Upsilon_{ri})$ become monotonically decreasing functions. By analyzing the relationship of the intersection, one can obtain that
(41)$$\alpha_{3}^{\prime }=\left\{\begin{array}{cc} \; & \min\left(\alpha_{0},\alpha_{1r,2r}\right),\;\;\;\mathrm{when}\;\alpha_{\min,1}\geq \alpha_{\max,2}\\ & \min\left(\alpha_{0},\alpha_{\min,1}\right),\;\;\mathrm{when}\;\alpha_{\min,1}<\alpha_{\max,2} \end{array}\right.$$
where $\alpha_{\min,1}=\min\left(\alpha_{1r,r1},\alpha_{1r,r2},\alpha_{1r,ma}\right)$, $\alpha_{\max,2}=\max\left(\alpha_{2r,r1},\alpha_{2r,r2},\alpha_{2r,ma}\right)$.

The above analysis leads to the optimal power control parameter, which satisfies OP2 under Case 2 as
(42)$$\alpha_{c2}^{*}=\left\{\begin{array}{cc} \; & \alpha_{1}^{\prime },\;\;H_{1}=H_{2}\\ & \alpha_{2}^{\prime },\;\;H_{1}>H2\\ & \alpha_{3}^{\prime },\;\;H_{1}<H_{2}. \end{array}\right.$$

Combining the optimal power control parameters of Case 1 and Case 2, we have the following optimal power control solution:(43)$$\alpha^{*}\;=\;\left\{\begin{array}{cc} \; & \alpha_{c1}^{*},\mathrm{Case}\;1:|l_{2}{|}^{2}(Q\;-\;\sigma_{u,r}^{2}|l_{r}{|}^{2})\;-\;\eta \rho Q|l_{r}{|}^{2}{\left|h_{2}\right|}^{2}\leq 0\\ \; & \alpha_{c2}^{*},\mathrm{Case}\;2:|l_{2}{|}^{2}(Q\;-\;\sigma_{u,r}^{2}|l_{r}{|}^{2})\;-\;\eta \rho Q|l_{r}{|}^{2}{\left|h_{2}\right|}^{2}>0 \end{array}\right.$$

Next, we assume a fixed *t* and ρ to derive the optimal power allocation ratio using the derived optimal power control parameters and the peak power limit of the source nodes. The derived theorem is described in Theorem 1.

**Theorem** **1.**
*For fixed parameters t and ρ, the optimal power allocation ratio that satisfies OP1 is obtained as*

(44)
$$\left(P_{1}^{*},P_{2}^{*}\right)=\left\{\begin{array}{cc} \; & \left(P_{1,\max},P_{2,\max}\right),\;\;if\;A_{1}\leq \alpha^{*}\leq A_{2}\\ & \left(P_{1,max},\frac{(1-\alpha^{*})Q}{l_{2}^{2}}\right),\;\;if\;\alpha_{1}\leq \alpha^{*}\leq 1\\ & \left(\frac{\alpha^{*}Q}{l_{1}^{2}},P_{2,\max}\right),\;\;if\;0\leq \alpha^{*}\leq \alpha_{2}\\ & \left(\frac{\alpha^{*}Q}{l_{1}^{2}},\frac{(1-\alpha^{*})Q}{l_{2}^{2}}\right),\;ifA_{2}\leq \alpha^{*}\leq A_{1} \end{array}\right.$$


(45)
$$\begin{array}{c} P_{R}^{*}=\min\left(\eta \rho \left(\gamma_{\Sigma }^{*}+\sigma_{u,r}^{2}\right)\frac{t}{1-t},\frac{Q}{|l_{r}{|}^{2}}\right) \end{array}$$

*where $\alpha^{*}$ is the optimal power control parameter that maximizes the minimum cognitive user transmission rate without considering the source peak power limits, $A=\frac{|l_{1}{|}^{2}P_{1,\max}}{Q}$, $A_{2}=1-\frac{|l_{2}{|}^{2}P_{2,\max}}{Q}$, $\alpha_{1}=\max\left(A_{1},A_{2}\right)$, $\alpha_{2}=\min\left(A_{1},A_{2}\right)$, and $\gamma_{\Sigma }^{*}=|h_{1}{|}^{2}P_{1}^{*}+{\left|h_{2}\right|}^{2}P_{2}^{*}$.*


**Proof.** When a limit on the source’s peak power is not enforced, the source transmit power can be written as $P_{1}=\frac{\alpha^{*}Q}{|l_{1}{|}^{2}}$, $P_{2}=\frac{(1-\alpha^{*})Q}{|l_{2}{|}^{2}}$ with optimum value $\alpha^{*}$. With a peak power constraint, $P_{1}$ and $P_{2}$ are separated into four cases based on the value of $\alpha^{*}$ according to Equations (Equation 21)–(Equation 22):  
B1:when $A_{1}\leq \alpha^{*}\leq A_{2}$, $P_{1}=P_{1,\max}$, $P_{2}=P_{2,\max}$B2:when $\max\left(A_{1},A_{2}\right)\leq \alpha^{*}\leq 1$, $P_{1}=P_{1,\max}$, $P_{2}=\frac{(1-\alpha^{*})Q}{|l_{2}{|}^{2}}$B3:when $0\leq \alpha^{*}\leq \min\left(A_{1},A_{2}\right)$, $P_{1}=\frac{\alpha^{*}Q}{|l_{1}{|}^{2}}$, $P_{2}=P_{2,\max}$B4:when $A_{2}\leq \alpha^{*}\leq A_{1}$, $P_{1}=\frac{\alpha^{*}Q}{|l_{1}{|}^{2}}$, $P_{2}=\frac{(1-\alpha^{*})Q}{|l_{2}{|}^{2}}$.Thus, if $\alpha^{*}\in \left\{B1,B2,B3,B4\right\}$, then the power control for such cases is guaranteed. Further, by substituting the obtained $P_{1}$ and $P_{2}$ into Equation (Equation 23), $P_{R}$ can be derived as shown in Equation (Equation 45).    □

The power allocation algorithm with fixed *t* and ρ is given in Algorithm 1.
**Algorithm 1** Power allocation under power control with fixed *t* and ρ.1:compute $F=|l_{2}{|}^{2}(Q-\sigma_{u,r}^{2}|l_{r}{|}^{2})-\eta \rho Q|l_{r}{|}^{2}{\left|h_{2}\right|}^{2}$, $\alpha_{0}$2:if F≤0, denote $P_{R}=Q/{\left|l_{r}\right|}^{2}$3:compute $\alpha_{ir,ma},i=1,2$, $\alpha_{1r,2r}$4:   if $\alpha_{1r,ma}\geq \alpha_{2r,ma}$5:     $\alpha^{*}=\alpha_{1r,2r}$6:   else if $\alpha_{1r,ma}<\alpha_{2r,ma}$7:       if $H_{1}>H_{2}$8:        $\alpha^{*}=\alpha_{1r,ma}$9:       else if $H_{1}\leq H_{2}$10:        $\alpha^{*}=\alpha_{2r,ma}$11:  end12:else if F>0, denote $P_{R}=\eta \rho \left(\left(\frac{|h_{1}{|}^{2}\alpha }{|l_{1}{|}^{2}}+\frac{|h_{2}{|}^{2}\left(1-\alpha \right)}{|l_{2}{|}^{2}}\right)Q+\sigma_{u,r}^{2}\right)\frac{t}{1-t}$13:compute $\alpha_{ir,ma}$, $\alpha_{1r,2r}$, $\alpha_{ir,rj}$, $\alpha_{\min,1}$, $\alpha_{\max,2}$14:    if $H_{1}=H2$15:      $\alpha^{*}=\min\left(\alpha_{0},\alpha_{1r,2r}\right)$16:    else if $H_{1}>H2$17:       if $\alpha_{\min,1}\geq \alpha_{\max,2}$18:        $\alpha^{*}=\min\left(\alpha_{0},\alpha_{1r,2r}\right)$19:       else if $\alpha_{\min,1}<\alpha_{\max,2}$20:        $\alpha^{*}=\min\left(\alpha_{0},\alpha_{\max,2}\right)$21:    else if $H_{1}<H2$22:       if $\alpha_{\min,1}\geq \alpha_{\max,2}$23:        $\alpha^{*}=\min\left(\alpha_{0},\alpha_{1r,2r}\right)$24:       else if $\alpha_{\min,1}<\alpha_{\max,2}$25:        $\alpha^{*}=\min\left(\alpha_{0},\alpha_{\min,1}\right)$26:  end27:end28:Then substitute $\alpha^{*}$ into Equations (Equation 44)–(Equation 45) to derive $P_{1}$, $P_{2}$, $P_{R}$

### 3.2. Optimal JoTAPS Scheme

This subsection derives the jointly optimum time allocation and power splitting ratio (JoTAPS) with a fixed transmit power. Firstly, we give an initial power control parameter $\alpha =\hat{\alpha }$. Then, the source nodes can determine the transmit power according to Equations (Equation 21) and (Equation 22), which are $P_{1}=\min\left(P_{1,\max},\frac{\hat{\alpha }Q}{|l_{1}{|}^{2}}\right)$ for $S_{1}$ and $P_{2}=\min\left(P_{2,\max},\frac{(1-\hat{\alpha })Q}{|l_{2}{|}^{2}}\right)$ for $S_{2}$. In this case, the previous optimization problem transforms into
(46)$$\begin{array}{cc} \; & \mathrm{OP}3:\;\;\max\;\min(R_{1},R_{2})\\ & s.t.\;\;C2:\;\mathrm{Equation}\;(15),\;C3:\;\mathrm{Equation}\;(9),\;C4:\;\mathrm{Equation}\;(17),\;C5,\;C6 \end{array}$$

By combining Equations (Equation 15) and (Equation 17), the rate region of the BC phase and the relay transmit power limit can be rewritten as
(47)$$\begin{array}{cc} \; & R_{i}\leq \left(1-t\right)\cdot C\left(\Upsilon_{ri}\right) \end{array}$$
(48)$$\begin{array}{cc} \; & \tilde{C2}:\eta \rho ({\left|h_{1}\right|}^{2}P_{1}+{\left|h_{2}\right|}^{2}P_{2}+\sigma_{u,r}^{2})\frac{t}{1-t}\leq \frac{Q}{|l_{r}{|}^{2}} \end{array}$$
where
(49)$$C\left({\hat{\Upsilon }}_{ri}\right)=\log_{2}\left(1+\frac{|h_{i}{|}^{2}\eta \rho \left(\gamma_{\Sigma }+\sigma_{u,r}^{2}\right)\frac{t}{1-t}}{\sigma_{u,i}^{2}+\sigma_{i}^{2}}\right)$$

Define $R_{\min}=\min\left(R_{1},R_{2}\right)$. The constraint of rate region C3 and C4 can be rewritten as
(50)$$\begin{array}{cc} \; & \tilde{C3:}\;R_{\min}\leq t\cdot g_{1}\left(t,\rho \right) \end{array}$$
(51)$$\begin{array}{c} \tilde{C4:}\;R_{\min}\leq \left(1-t\right)\cdot g_{2}\left(t,\rho \right) \end{array}$$
where
(52)$$\begin{array}{c} g_{1}\left(t,\rho \right)=\min\left(C\left(\Upsilon_{1r}\right),C\left(\Upsilon_{2r}\right),\frac{1}{2}C\left(\Upsilon_{max}\right)\right) \end{array}$$
(53)$$\begin{array}{c} g_{2}\left(t,\rho \right)=\min\left(C\left({\hat{\Upsilon }}_{r1}\right),C\left({\hat{\Upsilon }}_{r2}\right)\right) \end{array}$$

Substituting the rewritten constraint in Equations (48), (Equation 50) and (Equation 51) into OP3, we have
(54)$$\begin{array}{cc} \; & \mathrm{OP}4:\;\;\max_{t,\rho }\;R_{\min}\\ & s.t.\;\;\tilde{C2},\tilde{C3},\tilde{C4},C5,C6 \end{array}$$

Some analysis reveals that with respect to ρ (fix t) or with respect to *t* (fix ρ), OP4 is a convex optimization problem.

**Theorem** **2.**
*Given $\alpha =\hat{\alpha }$, OP4 is a convex optimization problem with respect to ρ (fix t) or with respect to t (fix ρ).*


**Proof.** To prove that OP4 is a convex optimization problem, we need to prove that the objective function and the constraint are both convex or affine functions. The objective function of OP4 is a constant, and when *t* and ρ are fixed, $\tilde{C2}$ is a linear function. Then, we derive the convex function properties of $\tilde{C3}$ and $\tilde{C4}$.When *t* is fixed, the first and second derivatives of $C\left(\Upsilon_{a}\right),a=\left\{1r,2r,MA\right\}$ in $g_{1}\left(t,\rho \right)$ with respect to ρ are
(55)$$\begin{array}{cc} \; & \frac{\partial C(\Upsilon_{a})}{\partial \rho }=-\frac{\gamma_{a}\sigma_{b}^{2}}{\ln2}\cdot \frac{1}{\left(\bar{\rho }\sigma_{u}^{2}+\sigma_{b}^{2}\right)\left(\bar{\rho }m_{a}+\sigma_{b}^{2}\right)} \end{array}$$
(56)$$\begin{array}{c} \frac{\partial^{2}C\left(\Upsilon_{a}\right)}{\partial^{2}\rho }=-\frac{\gamma_{a}\sigma_{b}^{2}}{\ln2}\cdot \frac{\sigma_{u}^{2}\left(\bar{\rho }\sigma_{u}^{2}+\sigma_{b}^{2}\right)+m_{a}\left(\bar{\rho }\sigma_{u}^{2}+\sigma_{b}^{2}\right)}{\left(\bar{\rho }\sigma_{u}^{2}+\sigma_{b}^{2}\right)\left(\bar{\rho }m_{a}+\sigma_{b}^{2}\right)} \end{array}$$
where $\bar{\rho }=1-\rho$, $m_{a}=\sigma_{u}^{2}+\gamma_{a}$, $\gamma_{a}=\left\{\gamma_{1},\gamma_{2},\gamma_{\Sigma }\right\}$ are three cases of a∈{1,2,Σ}.Since *t* is fixed, ρ∈(0,1). Thus $t\cdot \frac{\partial^{2}C\left(\Upsilon_{1r}\right)}{\partial \rho^{2}}<0$, $t\cdot \frac{\partial^{2}C\left(\Upsilon_{2r}\right)}{\partial \rho^{2}}<0$, $\frac{t}{2}\cdot \frac{\partial^{2}C\left(\Upsilon_{ma}\right)}{\partial \rho^{2}}<0$. From the properties of convex functions, we conclude that if f(x) and g(x) are convex (or concave) functions, then min(f(x),g(x)) are convex (or concave) as well. Clearly $g_{1}\left(t,\rho \right)$ is concave with respect to ρ when *t* is fixed; thus, $\tilde{C3}$ is concave with respect to ρ.When *t* is fixed, the first and second derivatives of $C\left(\Upsilon_{b}\right),b=\left\{r1,r2\right\}$ in $g_{2}\left(t,\rho \right)$ with respect to ρ are
(57)$$\begin{array}{cc} \; & \frac{\partial C({\hat{\Upsilon }}_{ri})}{\partial \rho }=\frac{|h_{1}{|}^{2}\eta m_{\Sigma }t\left(1-t\right)}{\ln2(|h_{1}{|}^{2}\eta \rho m_{\Sigma }t+m_{i})} \end{array}$$
(58)$$\begin{array}{c} \frac{\partial^{2}C\left({\hat{\Upsilon }}_{ri}\right)}{\partial \rho^{2}}=\frac{\left(\right|h_{1}{{|}^{2}\eta m_{\Sigma }t)}^{2}\left(1-t\right)}{\ln2[(|h_{1}{|}^{2}\eta \rho m_{\Sigma }t+m_{i}{)]}^{2}} \end{array}$$
where $m_{\Sigma =\sigma_{u}^{2}+\gamma_{\Sigma }}$, $m_{i}=\sigma_{u}^{2}+\sigma_{i}^{2},i=\left\{1,2\right\}$.Since *t* is fixed, ρ∈(0,1), $\frac{\partial^{2}C\left(\Upsilon_{ri}\right)}{\partial \rho^{2}}<0$. Thus, we can conclude that $g_{2}\left(t,\rho \right)$ is concave with respect to ρ, and so is $\tilde{C4}$ with respect to ρ.Now, we analyze the case when ρ is fixed. In this case, $t\cdot g_{1}\left(t,\rho \right)$ is a linear function. The second derivative of $\left(1-t\right)\cdot g_{1}\left(t,\rho \right)$ is
(59)$$\begin{array}{c} \frac{\partial^{2}C\left({\hat{\Upsilon }}_{ri}\right)}{\partial t^{2}}=-\frac{b^{2}}{\ln2\left(1-t\right){\left(1-t+bt\right)}^{2}}<0 \end{array}$$
where $b=\frac{|h_{i}{|}^{2}\eta \rho \left(\gamma_{\Sigma }+\sigma_{u}^{2}\right)}{\sigma_{u}^{2}+\sigma_{i}^{2}}$.Thus, OP4 is a concave function with respect to ρ (when fixing *t*) or with respect to *t* (when fixing ρ). Theorem 2 is proved.    □

Based on Theorem 2, we propose an alternating iterative optimization algorithm to determine optimal value of *t* and ρ. The first step of the algorithm is to solve for the optimal ρ with a given *t*. Let *k* be the iteration number. We can calculate the optimal value ρ of the k+1 iteration by solving OP5-a, which is written as
(60)$$\begin{array}{cc} \; & \mathrm{OP}5-a:\;\;\;\max\;\;R_{\min,a}\\ & s.t.\;\;\tilde{C2}:\;\;\rho^{\left(k+1\right)}\leq \frac{Q}{|l_{r}{|}^{2}}\cdot \frac{1-t^{\left(k\right)}}{\eta \left(\gamma_{\Sigma }+\sigma_{u,r}^{2}\right)t^{\left(k\right)}}\\ & \;\;\;\;\;\;\tilde{C3}:\;\;R_{\min,a}\leq t^{\left(k\right)}\cdot g_{1}\left(t^{\left(k\right)},\rho^{\left(k+1\right)}\right)\\ & \;\;\;\;\;\;\tilde{C4}:\;\;R_{\min,a}\leq \left(1-t^{\left(k\right)}\right)\cdot g_{2}\left(t^{\left(k\right)},\rho^{\left(k+1\right)}\right)\\ & \;\;\;\;\;\;C6:0<\rho^{\left(k+1\right)}<1. \end{array}$$

Then, by substituting the optimal value $\rho^{\left(k+1\right)}$ of the *k*th iterations into OP5, the optimal value $t^{\left(k+1\right)}$ of the *k*th iterations can be calculated by solving OP5-b, which is written as
(61)$$\begin{array}{cc} \; & \mathrm{OP}5-b:\;\;\;\;\max\;\;R_{\min,b}\\ & s.t.\;\;\tilde{C2}:t^{\left(k+1\right)}\leq \frac{Q}{Q+|l_{r}{|}^{2}\eta \left(\gamma_{\Sigma }+\sigma_{u,r}^{2}\right)\rho^{\left(k+1\right)}}\\ & \;\;\;\;\;\tilde{C3}:R_{\min,b}\leq t^{\left(k+1\right)}\cdot g_{1}\left(t^{\left(k+1\right)},\rho^{\left(k+1\right)}\right)\\ & \;\;\;\;\;\tilde{C4}:R_{\min,b}\leq \left(1-t^{\left(k+1\right)}\right)\cdot g_{2}\left(t^{\left(k+1\right)},\rho^{\left(k+1\right)}\right)\\ & \;\;\;\;\;C5:0<t^{\left(k+1\right)}<1. \end{array}$$

Since OP5-a and OP5-b are convex optimization problems, the value of $\rho^{\left(k+1\right)}$ and $t^{\left(k+1\right)}$ in each iteration can be solved by using CVX toolbox. With the solved $\rho^{\left(k+1\right)}$ and $t^{\left(k+1\right)}$, an alternative algorithm named jointly-optimum time allocation and power splitting (JoTAPS) is designed as shown in Algorithm 2, where ε is the given allowable deviation.
**Algorithm 2** The jointly-optimum time and power splitting scheme with fixed power allocation algorithm1:Initial $t=\frac{1}{2}$, $R_{\min}^{pre}=0$, k=12:While $|R_{\min}^{cur}-R_{\min}^{pre}|>\varepsilon$3:Solve OP5-a, and obtain $\rho^{\left(k+1\right)}$, $R_{\min,a}$4:Solve OP5-b, and obtain $t^{\left(k+1\right)}$, $R_{\min,b}$5:Update $R_{\min}^{cur}=R_{\min,b}$, k=k+16:End

### 3.3. Joint Power Control and Resource Allocation

Based on Algorithms 1 and 2, a stepped alternative optimization algorithm to solve OP1 is proposed. The idea of the algorithm is as follows:

Step 1: Give an initial power allocation ratio.

Step 2: Utilize Algorithm 2 to obtain the optimal power splitting ratio $\rho^{\left(k+1\right)}$ and time allocation ratio $t^{\left(k+1\right)}$.

Step 3: Substitute the obtained $\rho^{\left(k+1\right)}$ and $t^{\left(k+1\right)}$ as the initial value of Algorithm 1 to obtain the power allocation ratio.

Finally, the optimized parameter value of OP1 is satisfied by solving the above three steps iteratively. The proposed algorithm is described below.
**Algorithm 3** Joint power control and resource allocation scheme1:Initial $P_{1}$, $P_{2}$2:While l=1:loop3:Utilize Algorithm 2 to solve $\rho^{\left(k+1\right)}$ and $t^{\left(k+1\right)}$4:With $\rho =\rho^{\left(k+1\right)}$ and $t=t^{\left(k+1\right)}$, utilize Algorithm 1 to solve $\alpha^{*,l}$, $P_{1}^{*,l}$, $P_{2}^{*,l}$, $P_{R}^{*,l}$5:Update $\hat{\alpha }=\alpha^{*,l}$6:Update l=l+17:End

## 4. Numerical Results and Discussions

In this section, we provide simulation results to evaluate the proposed joint power control and resource allocation (JPCRA) scheme. Based on the above derivation, we notice that the system parameters such as interference threshold, power and PU interference value have an effect on the max–min achievable rate $R_{\min}$. Thus, in the simulation part, we first simulate and analyze the effects of these parameters on $R_{\min}$ in Figure 2, Figure 3 and Figure 4. To better demonstrate the superiority of the proposed JPCRA scheme, we compare it with the end-to-end achievable rates in Figure 5 and compare with three traditional optimization schemes in Figure 6, Figure 7 and Figure 8. The three traditional optimization schemes are the joint optimal power splitting and power allocation with fix time (FT-JoPSPA) scheme, joint optimal time and power allocation with fix power splitting (FPS-JoTPA) scheme and joint optimal time and power splitting with fix power allocation (FPA-JoTPS) scheme.

### 4.1. Simulation Setup and Parameters

The effectiveness of our proposed scheme is evaluated through experimental simulations in which the full-band urban indoor communication environment is taken into account. Let $d_{i}$ be the distances between the source $S_{i}$ and relay *R*, let $d_{iu}$ be the distances between the source $S_{i}$ to primary user PU and let $d_{ru}$ be the distances between relay node to PU, respectively. We consider $h_{i}=d_{i}^{-m}$, $l_{i}=d_{iu}^{-m}$ and $l_{r}=d_{ru}^{-m}$ as the channel gains between source $S_{i}$ and relay *R*, between source $S_{i}$ and the PU and between the relay node and the PU, respectively, where m=2 is the pass loss exponent. The channel gains between two nodes are reciprocal. To simplify the analysis, we consider a case where the sources and the relay are on a straight line, with $d_{1}+d_{2}=10$ m and $d_{1u}=d_{2u}=5$ m. The interference effects from the PU to each secondary users are assumed to be the same, i.e., $\sigma_{u,r}^{2}=\sigma_{u,i}^{2}=\sigma_{u}^{2}$. The noise power is given as $\sigma^{2}=10^{-9}$ W. Since full-band communication is considered, the frequency band is normalized as B=1. The parameters used in the simulation part are listed in Table 1.

### 4.2. The Effects of Parameters on $R_{\min}$

The efficiency of the parameter settings in the proposed technique is investigated according to the following figures.

Figure 2 depicts the influence of the interference tolerance threshold on the max–min cognitive user achievable rate. The simulation parameters are $(d_{1},d_{2})=$ (3, 7) m, $P_{1,\max}=$ (20, 15) dBm, $P_{2,\max}=15$ dBm, $\sigma_{u}^{2}=\left\{10^{-6},10^{-4}\right\}$ W. It is observed that when *Q* increases to a certain value, $R_{\min}$ initially increases and eventually saturates because the transmission rate of the cognitive user is a monotonically increasing function of transmit power $P_{i}$ which is affected by *Q* and the peak power constraint. When *Q* is sufficiently small, the transmit power is mainly limited by *Q*. When *Q* increases to a certain value (larger than the peak power), the transmit power is determined mainly by the peak power. Thus, increasing *Q* cannot continuously improve the achievable sum-rate.

Figure 3 depicts the the max–min achievable rate versus the peak power of source $S_{1}$ when $(d_{1},d_{2})=$ (3, 7) m, Q= (5, 10) dBm, $P_{2,\max}=$ 15 dBm and $\sigma_{u}^{2}=\left\{10^{-6},10^{-4}\right\}$. The result shows that $R_{\min}$ increases and tends to saturate to a stable value as $P_{1,\max}$ increases to a certain value. This is because $R_{\min}$ is proportional to the user’s transmission power, and the cognitive user transmission power is affected by *Q* and $P_{i,\max},i=1,2$ at the same time. When $P_{1,\max}$ is small, the user’s transmit power is limited by $P_{1,\max}$. Thus, $R_{\min}$ increases as $P_{1,\max}$. As $P_{1,\max}$ increases to a certain value (larger than *Q*), however, the user’s transmit power is limited by *Q*. Thus, continuing to increase $P_{1,\max}$ will not change $R_{\min}$.

Figure 4 depicts the max–min achievable rate obtained by the proposed JPCRA scheme versus the interference power $\sigma_{u}^{2}$ caused by the primary user to cognitive user when $(d_{1},d_{2})=$ (3, 7) m, Q= (5, 10) dBm, $P_{1,\max}=$ (20, 15) dBm, $P_{2,\max}=$ 15 dBm. It can be seen that $R_{\min}$ decreases and approaches 0 as $\sigma_{u}^{2}$ increases, and the descend degree of JPCRA is more gradual. Thus, consideration of the interference from the primary network is necessary to design the cognitive energy harvesting system.

### 4.3. Performance Analysis with Comparative Schemes

The superiority of the proposed scheme with benchmark schemes is investigated according to the following figures.

Figure 5 compares the max–min achievable rate ($R_{\min}$) obtained by the proposed JPCRA scheme with the end-to-end rate (we use $R_{i},i=1,2$ to denote the transmission rate of source ($S_{i}$) obtained by the maximization of achievable sum-rate (MASR) scheme [28]. In the simulation, parameters are set as $d_{1}=$ {2, 3} m, $d_{ru}=$ {2, 3} m, $(P_{1,\max},P_{2,\max})=$ (15, 15) dBm and $\sigma_{u}^{2}=\left\{10^{-6},10^{-4}\right\}$ W. This figure clearly shows that $R_{2}$ increases as $d_{1}$ decreases while $R_{1}$ remains the same, and $R_{\min}$ also increases, because the system with MASR scheme will enhance the transmission ability of the high channel quality to maximize the achievable sum-rate. Thus, the MASR scheme cannot enhance the transmission ability of the poor channel, which may mean that the transmission links with a poor channel quality do not meet their transmission needs. The proposed scheme considers the transmission fairness, effectively alleviating the effects caused by channel asymmetry. Therefore, when the system values the enhancement of the worst performance of a user most in this system, it is better to choose the proposed JPCRA scheme, whereas when the system values the enhancement of the system achievable sum rate most, it is better to choose the MASR scheme.

Figure 6 depicts the max–min achievable rate of the proposed JPCRA scheme versus the interference threshold of the primary network, which is compared with three traditional transmission schemes. The parameters are set as $(d_{1},d_{2})=$ (3, 7) m, $P_{1,\max}$ = 20 dBm, $P_{2,\max}=$ 15 dBm, $\sigma_{u}^{2}=10^{-6}$ W. It is observed that as the interference threshold increases, the $R_{\min}$ of each scheme increases and saturates at a stable value. This is because the transmit power of the cognitive user is mainly constrained by power control when the interference threshold is small; as the interference threshold increases to a certain value, the transmit power of the cognitive user is mainly constrained by the peak power limits. In this figure, the peak power is a fixed parameter; thus, the performance saturates as the interference threshold increases to a certain value. The proposed JPCRA scheme outperforms the other three schemes in the whole range of *Q*. When the interference threshold is a high SNR value, it is possible to use the FPA-JoTPS scheme to replace JPCRA, and when the interference threshold is a low SNR value, it is possible to use the FT-JoPSPA scheme to replace JPCRA.

Figure 7 shows the max–min achievable rate versus peak transmit power limit $P_{1,\max}$. The proposed JPCRA scheme is compared with three traditional transmission schemes. The parameters are set as $(d_{1},d_{2})=$ (3, 7) m, Q=5 dBm, $P_{2,\max}$= 15 dBm and $\sigma_{u}^{2}=10^{-6}$ W. The proposed JPCRA scheme clearly outperforms the three other schemes. It is also observed that as $P_{1,\max}$ increases, the $R_{\min}$ of each schemes increases and saturates to a stable value. This is because the transmit power is curved by the PU threshold, which curves the transmission rates. In this figure, it can be seen that when the peak transmit power limit $P_{1,\max}$ is sufficient, the proposed JPCRA scheme is definitely a good choice, whereas when $P_{1,\max}$ is in the low-SNR range, it is possible to use the three other traditional schemes to replace JPCRA.

Figure 8 illustrates the max–min achievable rate versus the interference power $\sigma_{u}^{2}$ caused by the primary user to cognitive user. The parameters are set as $(d_{1},d_{2})=$ (3, 7) m, Q= 5 dBm, $P_{1,\max}$ = 20 dBm and $P_{2,\max}$ = 15 dBm. As $\sigma_{u}^{2}$ increases, $R_{\min}$ decreases and approaches 0. The proposed JPCRA scheme outperforms the other three schemes, and the smaller $\sigma_{u}^{2}$ is, the greater the performance gap, while a smaller $\sigma_{u}^{2}$ results in a greater impact of resource allocation. Considering the influence of $\sigma_{u}^{2}$ on system performance, the primary user with less influence on the secondary network interference should be selected for spectrum sharing when selecting spectrum sharing objects.

### 4.4. Results Discussion

In this study, we investigate the performance of the proposed JPCRA scheme from two aspects. Firstly, the impact of parameter settings on the max–min achievable rate $R_{\min}$ is investigated. We verify that the performance growth of $R_{\min}$ is curved by the interference threshold of PU and the maximum transmit power constraint that echoes the cognitive two-way relay network should guarantee the performance of the primary network primarily. Then, the performance of the JPCRA scheme is compared with the MASR scheme, showing that the JPCRA scheme is preferable to enhance the poor channel quality. Last but not least, the superiority of the JPCRA scheme is verified compared with three traditional optimization schemes (FT-JoPSPA, FPS-JoTPA, FPA-JoTPS), which shows that the proposed JPCRA scheme achieves high performance in the whole range of adjustable SNR (changing *Q*, or $P_{1,\max}$, or $\sigma_{u}^{2}$).

## 5. Conclusions

We have studied the power control and resource allocation problem in the energy harvesting cognitive two-way relay network using the PS protocol and DF relay forwarding protocol. By considering transmission fairness, a joint resource allocation problem under power control is established, which aims at maximizing the minimum transmission rates of the cognitive users. To solve the optimization problem, a stepped alternative optimization algorithm, named the JPCRA scheme, is proposed to obtain the optimal parameter values. Simulation results have shown that the proposed scheme can improve the transmission performance of the cognitive users with poor channel quality and have verified its superiority compared with three other traditional schemes.

Further, we intend to expand the proposed scheme in an indoor relay environment or unmanned aerial vehicle cognitive relay circumstance. Another interesting further research direction could be exploring deep learning for cognitive relay selection.

## Figures and Tables

**Figure 1 sensors-23-07620-f001:**
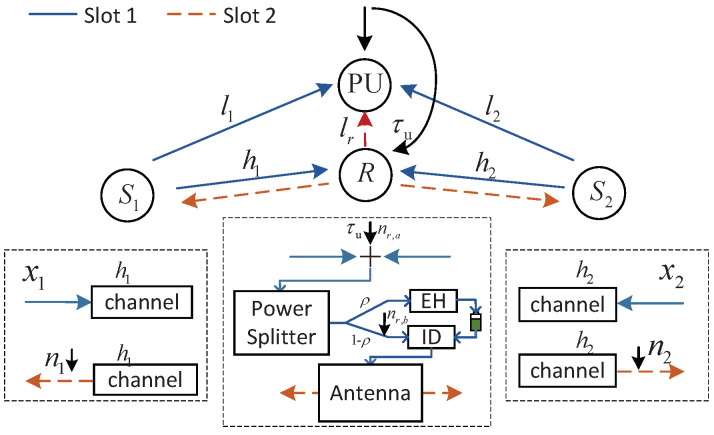
System model for two-way cognitive relay network.

**Figure 2 sensors-23-07620-f002:**
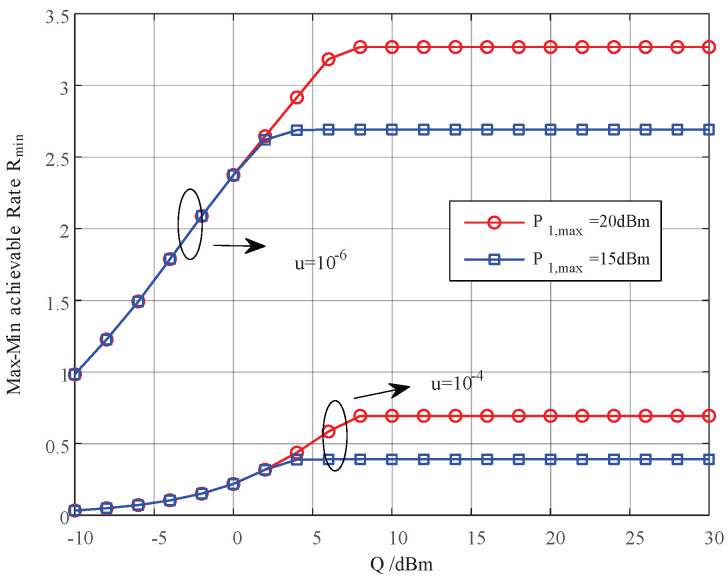
The max–min achievable rate vs. the interference threshold of PU.

**Figure 3 sensors-23-07620-f003:**
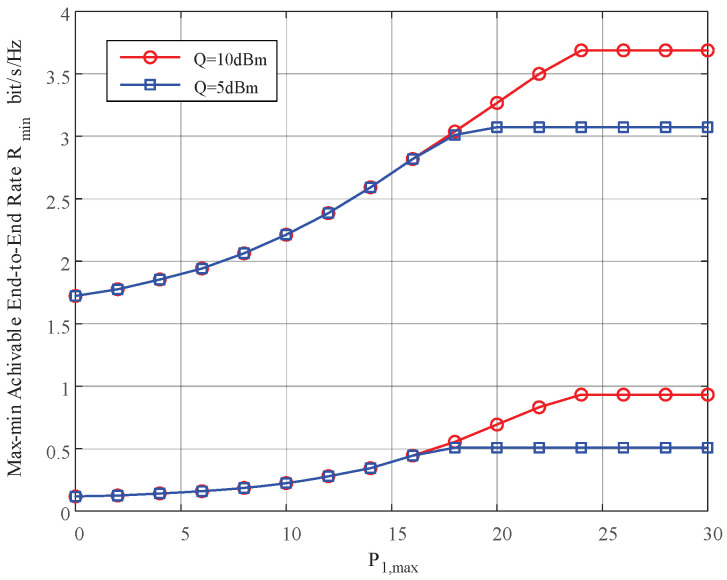
The max–min achievable rate vs. the maximum transmit power constraint of $S_{1}$.

**Figure 4 sensors-23-07620-f004:**
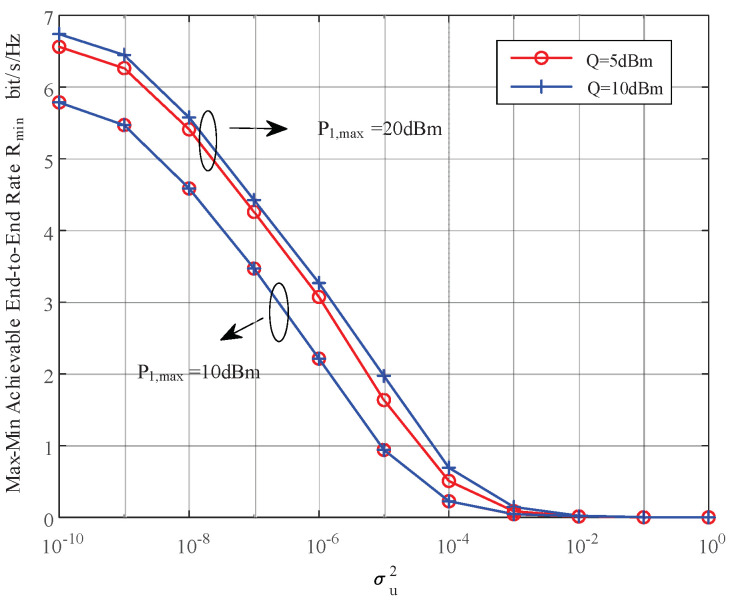
The max–min achievable rate vs. the interference to the cognitive network caused by PU.

**Figure 5 sensors-23-07620-f005:**
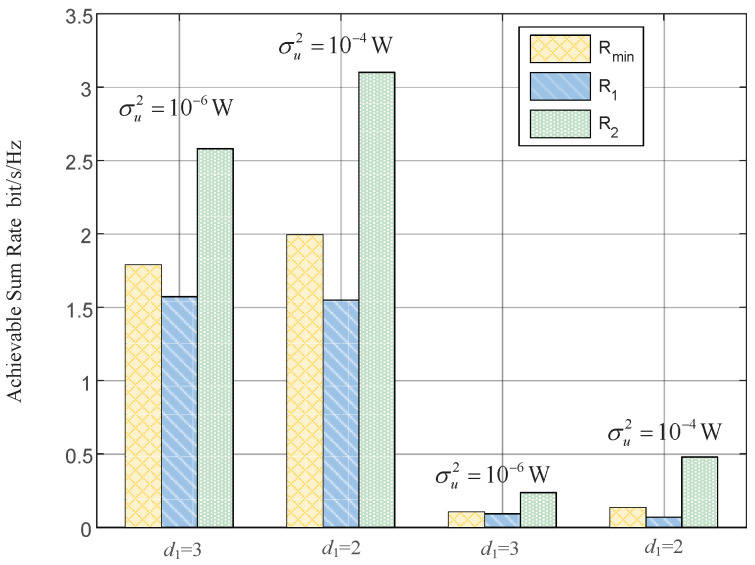
The max–min achievable rate vs. the end-to-end achievable rates aiming at maximizing the achievable sum rate.

**Figure 6 sensors-23-07620-f006:**
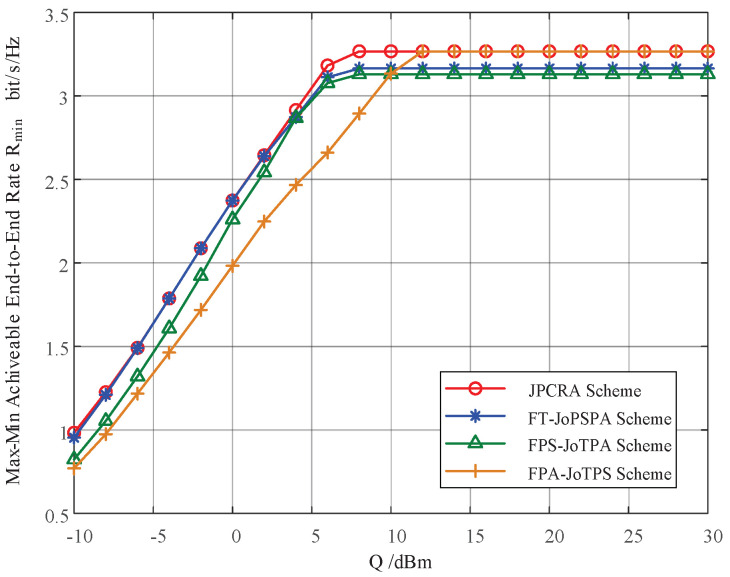
The max–min rates of different optimization schemes vs. the interference threshold of PU.

**Figure 7 sensors-23-07620-f007:**
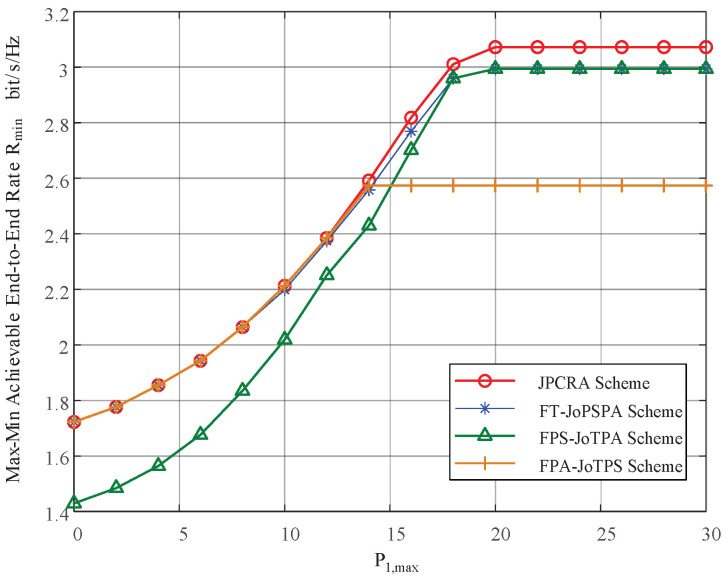
The max–min rates of different optimization schemes vs. the transmit power constraint of $S_{1}$.

**Figure 8 sensors-23-07620-f008:**
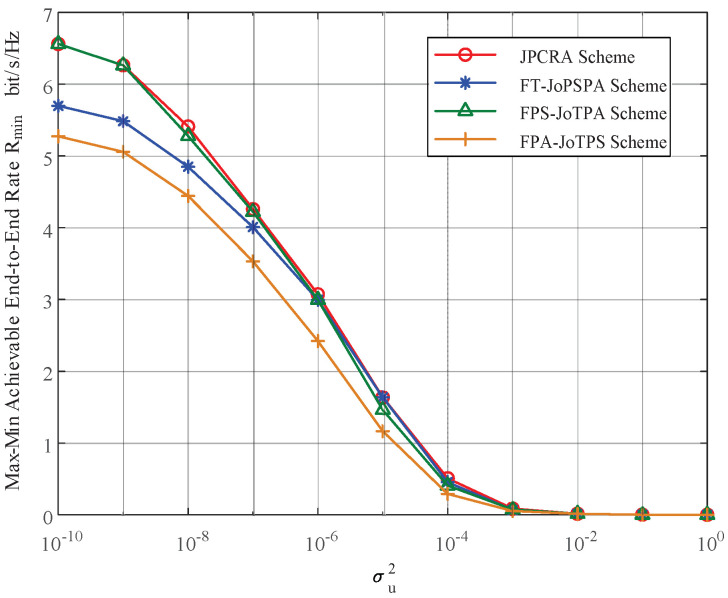
The max–min rates of different optimization schemes vs. the transmit power constraint of $S_{1}$.

**Table 1 sensors-23-07620-t001:** The parameters used in the simulation part.

Notation	Description	Value
$d_{i}$	The distance between source $S_{i}$ and relay *R*	0–10 m
$d_{iu}$	The distances between source $S_{i}$ and primary user PU	5 m
*m*	The pass loss exponent	2
$\sigma^{2}$	The white noise power at R	$10^{-9}$ W
$\sigma_{u}^{2}$	The interference caused by the primary network	($10^{-10}$, 1) W
$P_{i,max}$	The peak power of source node $S_{i}$	(0, 30) dBm
*Q*	The interference threshold	(−10, 30) dBm

## Data Availability

The data is unavailable due to privacy or ethical restrictions.

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
