# Peer review of "Joint Power Control and Resource Allocation with Rate Fairness Consideration for SWIPT-Based Cognitive Two-Way Relay Networks"

_sensors, 2023, doi:10.3390/s23177620_

Round 1

Reviewer 1 Report

The presented paper was devoted to the well-known issues related to the joint cognitive use of the spectrum and the acquisition of energy from the transmitted signal. The authors proposed their method by introducing corrections to already known solutions. During the simulation, three methods were referred to. In the initial part of the paper, a synthetic review of the references and achievements made in the field of research are presented. The analytical part explains in detail the issues of the conducted research. The reviewer believes that the article can be successfully published after preparation a few corrections and extensions:

1. Please briefly characterize all the methods that were referred to during the presentation of the simulation results. This will allow faster analysis of the presented results.

2. There is no chapter on the extensive discussion of the results obtained. Residual information on this subject can only be found in the chapter on simulation results.

3. It is necessary to comment on the specific purposefulness of using the proposed method, i.e. in what situations the proposed method is more effective than the ones already known. Some of these answers are contained in the results shown in the graphs.

4. In the presentation of the first simulation stage (lines 205-313), distance parameters are given that are not entirely clear. Please draw a diagram of the wireless connection network here, preferably in three-dimensional space, to reassure the reader of the physical structure of the network.

5. Many simplifications were made before starting the simulation. Please indicate what impact the real environment will have on the efficiency of the proposed method with the changing exponent m? Will the method be effective when directional antennas are introduced? What are the planned further directions of research that will prove the effectiveness and greater efficiency of the method in the presence of modern antenna systems?

6. The list of abbreviations placed at the end of the article should be deleted, and in the text it should be ensured that each abbreviation is expanded when it first appears, e.g. SWIPT, DF, EH.

7. Please review the text of the paper in terms of eliminating minor language errors and the so-called typos, for example:

a.       Rows 47-51 – please rebuild the description of the references.

b.       Formula (5) – please check indices at h variable

c.       Row 110 – the approximate equality sign is missing

d.       Row 171 – plural

e.       Please complete the missing units in the charts.

f.        Graphs can be placed side by side across the width of the page, greatly increasing the ability to perform visual analysis.

Some linguistic corrections are needed.

Author Response

Dear Reviewer,

Thank you so much for your comments that had led to an improved manuscript. We have revised the manuscript according to your comments. Please see the revised report in the uploaded reply word.

Sincerely,

Chunling Peng

Reviewer 2 Report

The paper proposes a new scheme for power control and resource allocation in SWIPT-based cognitive two-way networks. The authors first develop the mathematical/communication theory and then report some simulated results. The paper is sufficiently researched.

I have only few comments.

 1)    There is no mention of radio propagation data and issues (carrier frequency, bandwidth, multipath, antenna gain and pointing etc.), besides explicitly assuming Rayleigh fading in Section 4. The distances chosen (few meters) should be justified by mentioning real applications. A short Section dedicated to radio-propagation etc. should be written to guide readers.

2)    All abbreviations must be defined in the main text and reported in the list of Abbreviations.

3)     The same for the many mathematical symbols.

I suggest to include a final table in the Conclusion section summarizing advantages and and disadvantages of the four methods to clearly show why JPCRA should be preferred.

Author Response

Dear Reviewer,

Thank you so much for the comments that had led to an improved manuscript. We have revised the manuscript according to your comments. Please see the revised report in the uploaded reply word.

Sincerely,

Chunling Peng

Round 2

Reviewer 1 Report

Almost all the recommended corrections indicated in the previous version of the review were taken into account. Thank you for introducing corrections and extensions to the paper in accordance with the reviewer's recommendations. I have no further comments.

Author Response

Dear Revierer,

Thank you so much!

Sincerely,

Guozhong Wang

Reviewer 2 Report

The authors have practically not addressed any of the issues I have raised.

If propagation is not an issue, why mentioning Rayleigh fading? Propagation is fundamental in any wireless communications, therefore a section to it must be dedicated, including sensitivity to distance and to multipath. If these physical effects are not a problem, the authors shuld write a section explainging why it is so, not hiding this information in some other parts of the paper.

The authors do not consider the benefit of showing the list of abbreviations and mathematical symbols, and a table summarizing the overall merit of their method, because the paper will be longer! There is no page limitation in MDPI journals and I know that readers will benefit of these materials,

Author Response

Dear Reviewer,

Thank you so much for your comments. The responce to your comments is in the following reply letter.

Sincerely,

Guozhong Wang
